# Lipid-Lowering Drug Gemfibrozil Protects Mice from Tay-Sachs Disease via Peroxisome Proliferator-Activated Receptor α

**DOI:** 10.3390/cells12242791

**Published:** 2023-12-08

**Authors:** Sumita Raha, Debashis Dutta, Ramesh K. Paidi, Kalipada Pahan

**Affiliations:** 1Department of Neurological Sciences, Rush University Medical Center, Chicago, IL 60612, USA; sumitaraha@gmail.com (S.R.); duttad@musc.edu (D.D.); ramesh_kumar_paidi@rush.edu (R.K.P.); 2Division of Research and Development, Jesse Brown Veterans Affairs Medical Center, 820 South Damen Avenue, Chicago, IL 60612, USA

**Keywords:** Tay-Sachs disease, glial activation, GM2 ganglioside, gemfibrozil, peroxisome proliferator-activated receptor α

## Abstract

Tay-Sachs disease (TSD) is a progressive heritable neurodegenerative disorder characterized by the deficiency of the lysosomal β-hexosaminidase enzyme (Hex^−/−^) and the storage of GM2 ganglioside, as well as other related glycoconjugates. Along with motor difficulties, TSD patients also manifest a gradual loss of skills and behavioral problems, followed by early death. Unfortunately, there is no cure for TSD; however, research on treatments and therapeutic approaches is ongoing. This study underlines the importance of gemfibrozil (GFB), an FDA-approved lipid-lowering drug, in inhibiting the disease process in a transgenic mouse model of Tay-Sachs. Oral administration of GFB significantly suppressed glial activation and inflammation, while also reducing the accumulation of GM2 gangliosides/glycoconjugates in the motor cortex of Tay-Sachs mice. Furthermore, oral GFB improved behavioral performance and increased the life expectancy of Tay-Sachs mice. While investigating the mechanism, we found that oral administration of GFB increased the level of peroxisome proliferator-activated receptor α (PPARα) in the brain of Tay-Sachs mice, and that GFB remained unable to reduce glycoconjugates and improve behavior and survival in Tay-Sachs mice lacking PPARα. Our results indicate a beneficial function of GFB that employs a PPARα-dependent mechanism to halt the progression of TSD and increase longevity in Tay-Sachs mice.

## 1. Introduction

Tay-Sachs disease (TSD) disease is an autosomal recessive lysosomal storage disorder of the central nervous system (CNS), characterized by the heritable absence of the enzyme β-hexosaminidase A (HEXA), which is essential for the degradation of GM2 gangliosides. Therefore, the progressive accumulation of GM2 and subsequent neurodegeneration are pathological hallmarks of TSD [1,2]. As a rare disease, the carrier rate for TSD is about 1 in 250–300 people, with an incidence of approximately 1 in 320,000 live births in the US [2,3,4]. On the other hand, among Ashkenazi Jews, the incidence of TSD is estimated to be around 1 in 3600 [2,3,4].

GM2 ganglioside is an intermediate molecule of sphingolipids and other glycoconjugates. Therefore, mutations in the genes encoding *HEXA* cause the accumulation of GM2 ganglioside into the lysosome, leading to GM2 gangliosidoses Tay–Sachs (TSD, OMIM #272800). There are about 14 different axons of HEXA harboring more than 175 known mutations, including those found in French Canadians, Cajuns, Irish, and Brazilians [5]. Among Ashkenazi Jews, p.Tyr427Ilefs*5 and a donor splice-junction mutation in intron 12, c.1421+1G>C, account for over 90% of TSD cases, leading to severe, infantile-onset disease. On the other hand, among the non-Jewish population, while the c.1073+1G>A mutation results in severe disease, the p.Gly269Ser mutation is associated with adult-onset disease [2,5]. Sphingolipid metabolism, including gangliosides, is highly regulated for the differentiation and development of the central nervous system (CNS), and its expression is essential for the maintenance of the functional integrity of the nervous system [6,7]. The GM2 ganglioside accumulation leads to several cytotoxic effects that take place mainly in neurons, causing neuronal death [8].

Glial activation and dysfunction are salient features of neuroinflammatory and neurodegenerative diseases, such as Alzheimer’s disease (AD), Parkinson’s disease (PD), and multiple sclerosis (MS) [9,10,11,12]. Recent studies showed that the storage of GM1 and GM2 gangliosides in the CNS led to microgliosis and astrogliosis, and that the degree of inflammation correlates with increased levels of ganglioside accumulation [11]. Therefore, agents capable of inhibiting glial activation and inflammation in Tay-Sachs pathogenesis might offer neuroprotection in TSD. We previously revealed that gemfibrozil (GFB), a drug approved by the FDA for lipid lowering, inhibits the expression of inducible nitric oxide synthase (iNOS) and proinflammatory cytokines in astrocytes and microglia [13,14,15]. Here, we examined the effect of GFB on inflammation and overall glycoconjugate induced pathology in a mouse model of TSD, and observed that oral administration of GFB was capable of reducing glial inflammation and lowering glycoconjugates in the motor cortex of Tay-Sachs mice. Since GFB is an agonist of peroxisome proliferator-activated receptor α (PPARα), we investigated its role in GFB-mediated neuroprotection and found that GFB remained unable to reduce TSD pathology and increase longevity in Tay-Sachs mice lacking PPARα. Our results suggest that oral GFB may have therapeutic importance for TSD.

## 2. Materials and Methods

### 2.1. Reagents

Gemfibrozil (Figure 1) was purchased from Spectrum Chemical (New Brunswick, NJ, USA). Anti-Iba1 antibodies were purchased from Abcam (Cambridge, MA, USA). Anti-GFAP antibody (DAKO) was procured from Agilent (Santa Clara, CA USA), whereas anti-iNOS antibody was bought from BD Bioscience (San Jose, CA, USA). Alexa-fluor secondary antibodies used for immunofluorescence analyses were obtained from Jackson ImmunoResearch (West Grove, PA, USA), and IR-Dye-labeled secondary antibodies used for immunoblotting analyses were from Li-Cor Biosciences (Lincoln, NE, USA).

### 2.2. Animals

Tay-Sachs (B6;129S-*Hexa^tm1Rlp^*/J) mice or mice lacking *hexosaminidase a* or the *Hexa* gene were purchased from Jackson Laboratories. Experimental mice were housed under standard conditions with access to food and water ad libitum. Mice were bred and screened by genotyping. Tay-Sachs mice were also bred with PPARα knock out mice to generate bigenic Tay-Sachs^ΔPPARα^ mice. National Institutes of Health guidelines were followed for animal maintenance and experiments. Animal protocols were approved by the Institutional Animal Care and Use committee of Rush University Medical Center (Chicago, IL, USA).

### 2.3. Gemfibrozil (GFB) Treatment

GFB was solubilized in a 0.1% methyl cellulose solution. Tay-Sachs mice (3 months old) were treated with GFB at a dose of 8 mg/kg/day via gavage. Each mouse was fed with 100 µL of GFB solution by oral gavage daily for the next 60 days. Usually, any animal experiment is justified with a 99% confidence interval that generates *p* = 0.99 and (1 − *p*) = (1 − 0.99) = 0.01; ε is the margin of error = 0.05. Based on these values, the resultant sample size is: N = 1282 × 0.99 × (1 − 0.99) × 0.052 = 1282 × 0.99 × 0.01 × 0.052 = 0.016 × 0.0025 = 6.48. Therefore, six mice (*n* = 6) were used in each group.

### 2.4. Western Blotting

Immunoblotting was carried out as described previously [16]. After protein measurement, equal amounts of proteins were analyzed in 10% or 12% SDS-PAGE followed by transferring onto a nitrocellulose membrane. The blot was probed with primary antibodies overnight at 4 °C. The following primary antibodies [anti-iNOS (1:1000, BD Biosciences), anti-Iba1 (1:1000, Abcam), anti-GFAP (1:1000, Santa Cruz Biotechnology, Dallas, TX, USA), and anti-β-actin (1:5000, Abcam)] were used in this study. After overnight incubation, primary antibodies were removed, and the blots were washed with phosphate buffer saline containing 0.1% Tween-20 (PBST) and incubated with corresponding infrared fluorophore-tagged secondary antibodies (1:10,000, Jackson Immuno-Research) at room temperature. The blots were then incubated with secondary antibodies for 1 h. Later, blots were scanned with an Odyssey infrared scanner (Li-COR, Lincoln, NE, USA). ImageJ software (NIH, Bethesda, MD, USA) was used for quantification of band intensities.

### 2.5. Immunohistochemistry

Immunohistochemical analysis was carried out as mentioned before [16,17]. Briefly, mice were perfused transcardially with 4% paraformaldehyde, followed by keeping the brains in a 30% sucrose solution at 4 °C. Coronal sections (30 μm thickness) were cut from the forebrain containing striatum and motor cortex. Sections were blocked with 3% normal horse serum and 2% BSA made in PBST containing 0.5% Triton X-100 (Sigma-Aldrich, St. Louis, MI, USA) for 1 h. Sections were then kept in primary antibodies and incubated overnight at 4 °C temperature under shaking conditions. The next day, the samples were washed several times with PBST, followed by incubation with Cy2- or Cy5-labeled secondary antibodies (all 1:500; Jackson Immuno-Research) for 1 h under similar shaking conditions. Finally, after several washes with PBST, sections were incubated with 4′,6-diamidino-2-phenylindole (DAPI, 1:10,000; Sigma-Aldrich) for 5 min. Mean fluorescence intensity (MFI) was measured using Fiji (ImageJ2) [18,19].

### 2.6. PAS and H&E Staining

Periodic acid-Schiff (PAS) staining was performed with the PAS stain kit from Abcam (ab150680) as advised by the manufacturer. PAS-stained sections were counterstained with haematoxylin, dehydrated, cleared with a series of solutions of increasing concentrations of ethanol and xylene, and mounted in DPX (dibutyl phthalate xylene, British Drug Houses).

### 2.7. Open Field Test

Locomotor abilities of the animals were monitored by an open field test on a horizontal plane. A camera linked to the Noldus system and EthoVision XT software (The Netherlands) was employed to capture movement-associated parameters. The instrument records the overall movement abilities of the animals, such as total distance moved, velocity, moving time, resting time, center time, and frequencies of movement. Before recording the movement, all experimental mice were kept inside the open field arena for 10 min daily for 2 consecutive days for training and recording baseline values. After one day of rest, each mouse was gently placed in the middle of the open field arena, followed by data acquisition with the help of the software for 5 min [20].

### 2.8. Rotarod

It was performed as described before [17]. Briefly, mice were placed on the rotating rod against the direction of rotation. The machine was set to run at a gradually increasing speed of 4–40 rpm. The time for spending on the rotating rod was noted, and the experiment was ended once the animal slipped from the rod to the base of the instrument.

### 2.9. Gait Analysis

It was monitored as delineated earlier [21,22]. Briefly, mice were acclimatized by making them walk on a slanting platform for two consecutive days. Each mouse was given five trials each day to walk on the platform to the ascending direction. Mice were tested after one day of rest. In order to get the impression of the footprints of each animal, the gangway was covered with a long white paper, and the limbs of the animals were painted with non-toxic black ink. Different gait parameters, such as stride length, stride width, foot length, and toe spread, were recorded based on the footprints. If any animal stopped or started walking in reverse direction, the experiment for that animal was repeated.

### 2.10. Survival Assay

Another cohort of mice was sacrificed at the endpoint to assess survival analysis. All mice were followed daily for 6 months to record survival. Survival time reflects the time required for the animals to reach parameters of measurable endpoints, such as reduced gait and motor movements and paralysis of fore and hind limbs. Survival data were plotted using the Kaplan–Meier method, and different groups were compared using the Log–rank (Mantel–Cox) test (GraphPad Prism Software v.9.0).

### 2.11. Statistics

Statistics were performed using GraphPad Prism version 9.0. One-way ANOVA followed by Tukey’s multiple comparison test was performed for analyzing statistical significance among multiple samples, whereas unpaired two-tailed *t*-test was employed to compare two samples. Values are expressed as mean ± S.E.M., and the criterion for statistical significance was *p* < 0.05.

## 3. Results

### 3.1. Oral Administration of GFB Prevents Activation of Astrocytes and Microglia in the Motor Cortex of Tay-Sachs Mice

Neurodegenerative disorders are often characterized by the proliferation and activation of astrocytes and microglia, which tend to accelerate the disease process [23,24,25]. We, therefore, sought to investigate the effect of oral GFB on gliosis and inflammation. Immunostaining of astrocyte specific marker, glial fibrillary acidic protein (GFAP), exhibited that the Tay-Sachs mice had a significantly higher number of astroglia in the motor cortex region when compared to the non-Tg mice (Figure 2A,C). Moreover, the expression of the nitrosative stress marker inducible nitric oxide synthase (iNOS) was also found to be significantly upregulated in GFAP +ve astroglia in the motor cortex region of Tay-Sachs mice when compared to the non-Tg mice (Figure 2A,B,D). Western blot of GFAP and iNOS of these brain tissues corroborated these results (Figure 2E–G). Interestingly, the number of activated astroglia as well as the expression of astroglial iNOS decreased by GFB treatment in Tay-Sachs mice suggesting that GFB profoundly exerts an anti-inflammatory effect in TSD mouse model.

Next, we monitored microgliosis in the motor cortex region of Tay-Sachs mice and found increased immunostaining of microglia by its marker ionized calcium binding adaptor molecule 1 (Iba1) (Figure 3A,B). Furthermore, greater colocalization of iNOS with microglia was seen in Tay-Sachs mouse brains as compared to the non-Tg mice (Figure 3A,C). Moreover, immunoblot analysis confirmed a significant increase in the protein expression of Iba1 in Tay-Sachs mice as compared to the non-Tg group (Figure 3D,E). Interestingly, we found that consistent to the inhibition of astrogial activation, oral GFB significantly reduced microglial inflammation.

### 3.2. Treatment with GFB Attenuates Neuronal Apoptosis and Reduces Glycoconjugates in Tay-Sachs Mice

Apoptosis was detected by the in situ terminal deoxynucleotidyl transferase-mediated dUTP nick end labeling (TUNEL) method. We surveyed the motor cortex region of the brain in severely symptomatic Tay-Sachs mouse brain of 3–4 months of age. The number of apoptotic neurons was significantly higher in Tay-Sachs mice compared to the age-matched Non-Tg mice. Oral treatment with GFB significantly reduced neuronal apoptosis (Figure 4A,B).

We also used routine hematoxylin and eosin (H&E) stained sections to evaluate the tissues of the motor cortex region of the brain and then added periodic acid-Schiff (PAS) and immunohistochemistry (IHC) staining methods to characterize intricate details. Pathological amounts of GM2 ganglioside and related glycoconjugates, reported elevated as early as in fetal life, appear to increase linearly over time, and by age 4w are easily visible by PAS staining [26]. In the present study, we evaluated the motor cortex region of 3–4 month old Tay-Sachs mice stained with periodic acid-Schiff reagent (PAS) to detect stored glycoconjugates, a hallmark of GM2 gangliosides.

Immunohistochemistry revealed widely dispersed PAS positive granular staining in Tay-Sachs mice that was mostly cytoplasmic and restricted to macrophages and microglia when compared to the Non-Tg wild-type control. Moreover, Tay-Sachs mice also had markedly increased amounts of larger and paler staining granular material in the perinuclear cytoplasm of numerous large and medium-sized neurons (Figure 4C,D). Oral treatment with GFB significantly reduced PAS positive granules and had lower amounts of stained granular material present in smaller amounts and in far fewer neurons. The glycoconjugate material-stained magenta is shown by the arrow and quantified for PAS +ve granules per mm square (Figure 4C,D). Tay-Sachs mice compared to Non-Tg showed widely dispersed swollen dystrophic axons, neurites, and spheroids in multiple regions of the tissue. Moreover, there was severe vacuolation in Tay-Sachs mice compared to Non-Tg mice (Figure 4E,F). In marked contrast, the pathological findings observed in GFB-treated mice displayed reduced swollen dystrophic axons, neurites, and a significant reduction in the number of vacuoles (Figure 4E,F). Our finding indicates that oral administration of GFB activates the expression of β-hexosaminidase that clears the stored material and restores apparent normal histology.

### 3.3. Oral Administration of GFB Improves Gait, Alleviates Motor Deficits, and Increases Survival in Tay-Sachs Mice

The neuroprotective effect of GFB treatment against Tay-Sachs-associated brain pathology necessitated the evaluation of the behavioral parameters of the experimental mice. In the present investigation, our focus was on the motor cortex, the region known to regulate motor coordination and movement. To investigate motor coordination in more details, we performed footprint analysis to measure gait. While the stride length of Tay-Sachs mice was significantly lower than the Non-Tg animals, the toe spread increased (Figure 5A,B) when affected by disease pathology (Figure 5A,E). This indicates that Tay-Sachs mice, due to compromised movement capability, take a longer time to cross the gangway. We also measured stride width (Figure 5A,C) and foot length (Figure 5A,D) of the experimental animals. There was a significant increase in the stride width and length of the Tay-Sachs mice compared to the Non-Tg. Interestingly, after GFB treatment, all parameters, including stride length, toe spread, stride width, and foot length of Tay-Sachs mice, significantly showed values that are more towards the values of Non-Tg animals, thus showing improvement in overall motor skills (Figure 5A–E).

Additionally, we analyzed the motor behavior performance of the animals. As expected, Tay-Sachs mice showed extremely poor performance in the open field test, as shown by the reduction in the movement parameters such as distance moved (Figure 5F,G) and velocity (Figure 5F,H) of movement in the arena, which were significantly less when compared to the Non-Tg mice. In addition, the latency time taken by Tay-Sachs mice in rotarod (Figure 5I) indicates that coordination of feet movement is extremely impaired. In contrast, Tay-Sachs mice administered with GFB exhibited significantly improved performance in all the behavioral experiments, evident by higher velocity, larger distance moved, and better moving abilities in the open field arena and by a higher latency time taken by these animals in the rotarod test.

Survival curves for each treatment group were estimated using the Kaplan–Meier method. Oral gavage of gemfibrozil had a marked effect on the life span of Tay-Sachs mice, increasing the survival time from 110 days in untreated mice to 210 days in the treated mice, as shown by Kaplan–Meier survival curve analysis (Figure 5J).

### 3.4. Oral GFB Induces the Activation of PPARα in Tay-Sachs Mice

Previous reports from our group have shown PPARα to act as a master regulator for many neurodegenerative diseases. We therefore sought to validate the effect of GFB on basal levels of PPARα. The motor cortex region of Non-Tg, Tay-Sachs, and GFB-treated Tay-Sachs were double-labeled for PPARα and NeuN (Figure 6A,B). We found a significant decrease in neuronal PPARα in Tay-Sachs mice compared to the Non-Tg. There was a significant increase in neuronal PPARα in mice treated with GFB (Figure 6A,B). This was further validated using immunoblotting techniques, which further demonstrated a significant reduction of PPARα in Tay-Sachs mice compared to Non-Tg (Figure 6C,D). Oral GFB administration showed a substantial increase in the levels of neuronal PPARα (Figure 6A,B). Given the observation that GFB can activate PPARα, we explored the hypothesis that activation of PPARα could be the mechanism by which it exhibits any glycoconjugate-lowering effects. We wanted to further investigate whether PPARα had a role in reducing glycoconjugates in the brain of Tay-Sachs mice. Therefore, we generated Tay-Sachs^DPPARα^ (Tay-Sachs mice lacking PPARα) mice by breeding Tay-Sachs mice with PPARα knock-out mice (Figure 6F). In this regard, we used PAS staining to compare the glycoconjugate pathology among groups of mice. Our results indicate no difference in the PAS positive granules of Tay-Sachs and Tay-Sachs^ΔPPARα^ mice compared to the Non-Tg (Figure 6G). The glycoconjugate material-stained magenta was quantified for PAS +ve granules per mm square in all three groups of mice (Figure 6H).

### 3.5. GFB Attenuates Glycoconjugate Materials in Tay-Sachs Mice via PPARα

Given the observation that GFB can activate PPARα, we explored our hypothesis that activation of PPARα by GFB could be the underlying mechanism by which it exhibits the glycoconjugate-lowering effect. We compared the glycoconjugate pathology of the motor cortex region between all the groups, *viz*. Non-Tg, Tay-Sachs, Tay-Sachs-treated with GFB, Tay-Sachs^ΔPPARα^, and Tay-Sachs^ΔPPARα^ treated with GFB, using PAS staining followed by quantification of PAS +ve granules per mm square (Figure 7A,B). Hematoxylin staining followed by quantification of the number of vacuoles was done in the similar groups (Figure 7C,D). Histological observation using PAS stain revealed increased glycoconjugate storage material in both Tay-Sachs and Tay-Sachs^ΔPPARα^ when compared to normal Non-Tg mice (Figure 7A,B).

Treatment with GFB markedly showed clearance of storage in Tay-Sachs mice, but there was no alteration in the levels of storage material in GFB-treated Tay-Sachs^ΔPPARα^ mice (Figure 7A,B), thereby suggesting that activation of PPARα by GFB could be the underlying mechanism behind its glycosphingolipid-attenuating effects. It is noteworthy to mention that GFB-treated Tay-Sachs^ΔPPARα^ mice exhibited similar pathology as the Tay-Sachs mice, indicating that ablation of PPARα did not aggravate the accumulation of glycosphingolipids in these mice. Similarly, routine H&E analysis of the same groups revealed a significant increase in neuronal vacuolation in both Tay-Sachs and Tay-Sachs^ΔPPARα^ as compared to normal Non-Tg (Figure 7C,D). Histologic examination revealed neurons to be balloon-shaped with cytoplasmic vacuoles (Figure 7C,D). GFB-treated Tay-Sachs mice showed a significant reduction in vacuolation whereas there was no change in GFB-treated Tay-Sachs^ΔPPARα^ mice (Figure 7C,D). Together, these findings suggest that GFB treatment of Tay-Sachs mice could mitigate glycoconjugate pathology in a PPARα-dependent manner.

### 3.6. Oral Administration of GFB Improves Motor Deficits and Increases Survival in Tay-Sachs Mice via PPARα

Finally, we analyzed the effect of GFB on the behavioral performance of Non-Tg, Tay-Sachs, GFB-treated Tay-Sachs, Tay-Sachs^ΔPPARα^, and GFB-treated Tay-Sachs^ΔPPARα^ mice (Figure 8A–C). We examined the general locomotor activity of these mice in the open field test (Figure 8A), which showed significant differences in the total distance traveled (Figure 8B) and velocity (Figure 8C) of different cohorts of mice.

Tay-Sachs mice and Tay-Sachs^ΔPPARα^ mice demonstrated severe impairment in the open field test, as shown by a reduction in movement (Figure 8A). The Tay-Sachs and Tay-Sachs^ΔPPARα^ mice also exhibited poor performance in the rotarod test (Figure 8D), affirming compromised motor cortex parameters such as distance covered and mean velocity. In addition, the coordination and muscle strength experienced by these Tay-Sachs and Tay-Sachs^ΔPPARα^ mice were significantly reduced. However, GFB treatment reversed the poor motor performance and locomotor activities of the Tay-Sachs animals but not the Tay-Sachs^ΔPPARα^ mice, indicating that oral administration of GFB improves motor deficits in Tay-Sachs mice via PPARα.

Survival curves for each treatment group, *viz.* Non-Tg, Tay-Sachs, GFB-treated Tay-Sachs, Tay-Sachs^ΔPPARα^, and GFB-treated Tay-Sachs^ΔPPARα^ mice, were estimated. Interestingly, GFB-treated Tay-Sachs mice survived for more than 6 months, as shown by the Kaplan–Meier survival curve analysis (Figure 8E). However, GFB-treated Tay-Sachs^ΔPPARα^ mice failed to survive beyond 3 months (Figure 8E), suggesting an essential role of PPARα in GFB-mediated increased longevity of Tay-Sachs mice.

## 4. Discussion

Tay-Sachs disease (TSD), a fatal inherited lysosomal storage disorder, leads to neurological dysfunction of the brain primarily due to the lack of Hex A, an enzyme characterized by defective GM2 ganglioside [27]. Infants with TSD appear healthy at birth, but progressive GM2 accumulation causes loss of motor function and cognition, developmental regression, hind limb plasticity, muscle weakness, dystonia, blindness, seizures, and death in childhood [28]. Despite intensive investigations, there is no effective therapy or lines of treatment available for TSD. Although several therapeutic approaches have been suggested, like therapeutic enzyme replacement [29], bone marrow transplantation [30], substrate deprivation therapy [31], hematopoietic [32] or neural stem cell transplantation [33], use of chemical chaperones and oligonucleotide recombination [34], or a combination of all. However, until now, none of these therapies has led to a successful treatment option for TSD.

GFB, known as Lopid in the pharmacy, has been used in humans for the treatment of hyperlipidemia since its FDA approval in 1982 without any report of adverse incident [35]. Although continuous treatment of rodents with fibrate drugs like gemfibrozil for 45–50 weeks leads to the formation of hepatic tumor [36,37], such tumor promotion is not seen in humans, other primates, and guinea pigs, species that have lost their capacity to produce ascorbate due to inherent loss of the *gulonolactone oxidase* gene [36,38]. Several lines of indication presented in this manuscript clearly exhibit that GFB is capable of protecting mice from Tay-Sachs toxicity. Our conclusion is based on the following. *First*, similar to TSD, neuroinflammation driven by activated astroglia and microglia is seen in the brain of Hexa^−/−^ mice (here termed as Tay-Sachs mice). It is being increasingly recognized that understanding the heterogeneity of microglial activation in the context of disease may facilitate the design of therapeutics that dampen the detrimental effects of microglial activation [39,40]. Here, two months of GFB treatment markedly reduced astrocytic and microglial inflammation in the motor cortex of Tay-Sachs mice. *Second*, gait abnormality is a characteristic feature of TSD as well as Tay-Sachs mice. However, oral GFB significantly inhibited gait and behavioral impairments in Tay-Sachs mice. *Third*, as seen in TSD, Tay-Sachs mice also die early. However, a significant increase in life expectancy of Tay-Sachs mice was seen after GFB treatment. These results suggest that GFB may have implications in TSD therapy.

Due to the mutation of the *Hexa* gene, patients with Tay-Sachs disease exhibit the accumulation of G_M2_ ganglioside in the CNS, and therefore, successful treatments should be associated with the reduction of ganglioside deposition. While untreated Tay-Sachs mice displayed ganglioside inclusion bodies in the motor cortex, GFB treatment markedly reduced ganglioside deposition. It has been shown that the accumulation of undegraded gangliosides is directly linked to the activation of microglial cells [41,42]. It is nice to see that clearance of storage materials mapped directly to regions where enzymatic activity was present, as determined by PAS staining, and microglia expansion/activation was also absent in the same regions. Several studies have also delineated that pathological accumulation of GM1 or GM2 is capable of triggering apoptosis [8,43,44]. Accordingly, marked apoptosis was found in the motor cortex of Tay-Sachs mice that was strongly inhibited by oral GFB. Therefore, by suppressing pathological ganglioside accumulation, GFB treatment may decrease glial inflammation and protect neurons from apoptosis.

The mechanism by which GFB may reduce ganglioside accumulation from the brain is also becoming clear. Lysosomal biogenesis and associated autophagy play an important role in the removal of aggregated biomolecules from the cells. Several studies from our lab and others have shown that GFB stimulates lysosomal biogenesis and autophagy in neurons and glial cells [16,45,46]. Transcription factor EB (TFEB) is considered the master regulator of lysosomal biogenesis, and we have also seen that GFB increases lysosomal biogenesis via peroxisome proliferator-activated receptor α (PPARα)-mediated transcription of TFEB [45]. Moreover, activation of PPARα is reported to reduce neuroinflammation via upregulation of anti-inflammatory molecules such as SOCS3 and IL-1Ra [47,48]. Hence, to find out the core mechanism behind ganglioside reduction, we primarily sought to monitor the level of PPARα in the brains of Tay-Sachs mice. Therefore, reduced levels of PPARα in the motor cortex of Tay-Sachs mice might be responsible for the chronic accumulation of GM2 gangliosides. This finding was verified in Tay-Sachs^ΔPPARα^ mice in which GFB remained unable to reduce the accumulation of PAS-stained glycoconjugates and improve longevity. Therefore, it appears that GFB-mediated activation of PPARα is perhaps the underlying reason behind the neuroprotective effect of GFB in Tay-Sachs mice.

Although currently, there is no approved therapy or cure for TSD, scientists are proposing several approaches (e.g., enzyme replacement therapy, substrate reduction therapy, hematopoietic stem cell transplantation, gene therapy, chemical chaperones, antisense oligonucleotides, etc.) [29,30,31,32,33,34]. However, GFB has several advantages over these proposed approaches. For example, GFB can be taken orally, and it is fairly non-toxic [14]. After oral intake, it is capable of crossing the blood-brain barrier [48]. It increases the lifespan of *Cln2^(−/−)^* mice, an animal model of late-infantile neuronal ceroid lipofuscinosis, via PPARα [49]. However, at present, we do not know anything about the level of PPARα in TSD patients. Since GFB needs PPARα to display neuroprotection in Tay-Sachs mice, in the absence of a basal level of PPARα, GFB may not exhibit optimal therapeutic efficacy in TSD patients. Therefore, future studies could be directed to address this aspect.

## 5. Conclusions

In summary, this study demonstrates that the oral administration of gemfibrozil, an FDA-approved lipid-lowering drug in humans, decreases glial inflammation, reduces glycoconjugate storage materials, attenuates apoptosis, increases locomotor activities, and extends the lifespan of Tay-Sachs mice. Mechanistically, GFB exhibits a neuroprotective effect in Tay-Sachs mice via PPARα. Although the in vivo state of Tay-Sachs mice does not truly recapitulate the in vivo scenario of TSD patients, and not much is known about the status of PPARα in TSD, our results suggest that oral GFB may have therapeutic importance in TSD.

## Figures and Tables

**Figure 1 cells-12-02791-f001:**
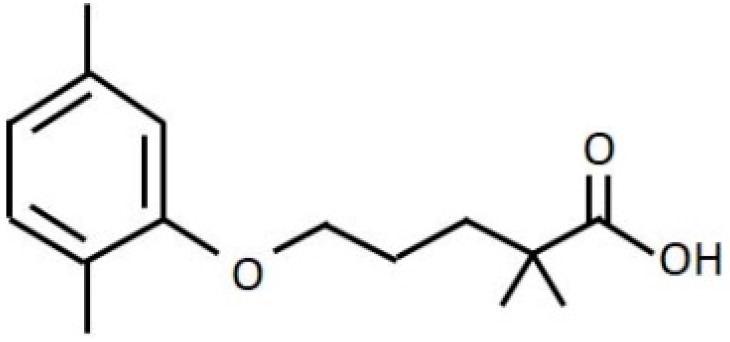
Structure of gemfibrozil (GFB).

**Figure 2 cells-12-02791-f002:**
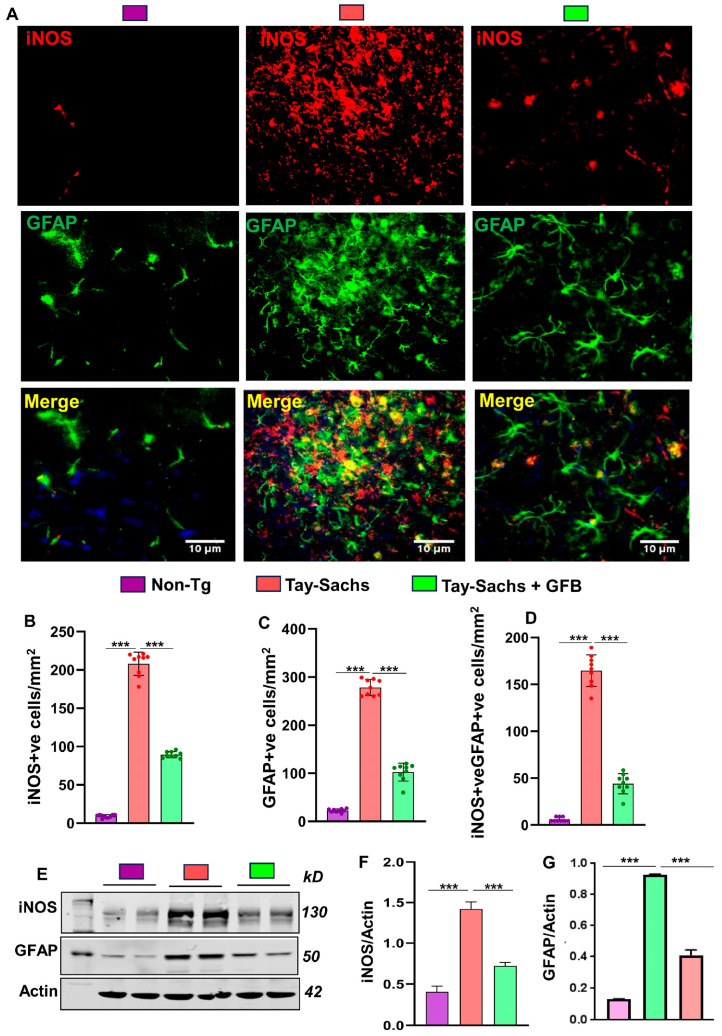
Oral administration of GFB mitigates the activation of astrocytes in the cerebral cortex of Tay-Sachs mice. Three-month-old Tay-Sachs mice (*n* = 6/group) were treated with GFB (8 mg/kg/d) solubilized in 0.5% methylcellulose. Therefore, the control group of Tay-Sachs mice received 0.5% methylcellulose as a vehicle. After 2 months of daily treatment, mice were sacrificed. Age-matched non-transgenic mice (*n* = 6/group) were used as control. Astroglial activation was monitored in cerebral cortex sections by double-label immunofluorescence for GFAP and iNOS (**A**) followed by quantification of iNOS +ve (**B**) and GFAP +ve cells (**C**) and iNOS +ve, GFAP +ve cells (**D**). Results represent counting of three different sections from three different mice (n = 3) per group with ImageJ software. Cerebral cortex homogenates (*n* = 4 per group) were subjected to immunoblot analysis for GFAP using β-actin as a loading control (**E**). Densitometric analysis of relative iNOS (iNOS/Actin) (**F**) and GFAP (GFAP/Actin) (**G**) levels with respect to non-transgenic was measured with ImageJ. All data represent mean ± SEM. Statistical analyses were performed by one-way ANOVA, followed by Dunnett’s multiple comparison test; *** *p* < 0.001. Label color code: top right; scale bars are as marked. GFAP, Glial fibrillary acidic protein; iNOS, inducible nitric oxide synthase.

**Figure 3 cells-12-02791-f003:**
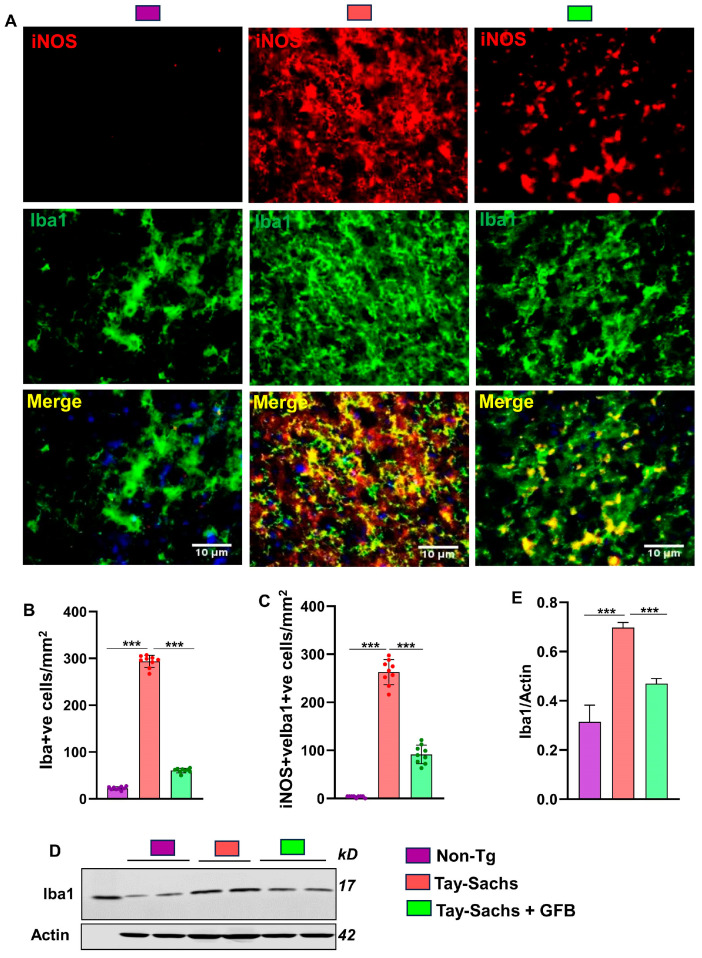
Oral administration of GFB inhibits the activation of microglia in vivo in the cerebral cortex of Tay-Sachs mice. Three-month-old TS mice (*n* = 6/group) were treated via oral gavage with GFB (8 mg/kg/d) solubilized in 0.5% methylcellulose. Therefore, control TS mice received 100 µL 0.5% methylcellulose as a vehicle. After 2 months of daily treatment, mice were sacrificed. Age-matched non-transgenic mice (*n* = 6/group) were used as control. Microglial activation was monitored in brain (cerebral cortex) sections by double-labelled immunofluorescence for Iba1 and iNOS (**A**) followed by quantification of Iba1 +ve (**B**) and iNOS +ve, Iba1 +ve cells (**C**). Results represent counting of three different sections from three different mice (*n* = 3) per group with ImageJ software. Cerebral cortex homogenates (*n* = 4 per group) were subjected to immunoblot analysis for Iba1 using β-actin as a loading control (**D**). Densitometric analysis of relative Iba1 (Iba1/Actin) (**E**) levels with respect to non-transgenic was measured. All data represent mean ± SEM. Statistical analyses were performed by one-way ANOVA, followed by Dunnett’s multiple comparison test; *** *p* < 0.001. Iba1, Ionized calcium binding adapter molecule 1; iNOS, inducible nitric oxide synthase.

**Figure 4 cells-12-02791-f004:**
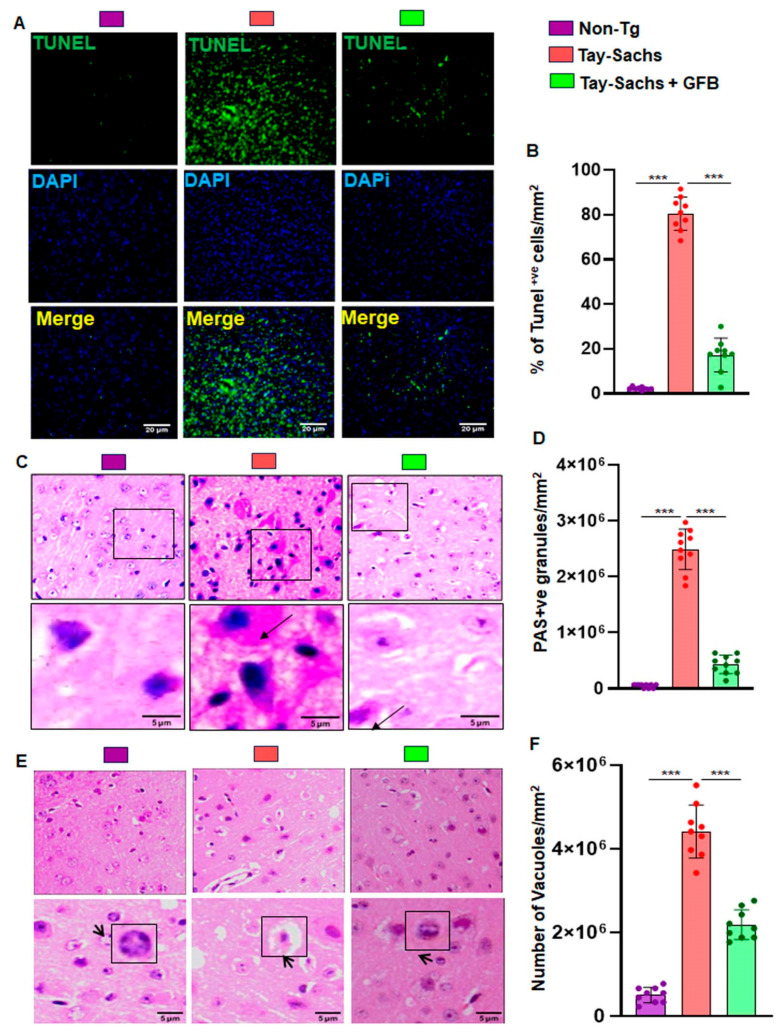
Treatment with GFB attenuates neuronal apoptosis and reduces glycoconjugates in Tay-Sachs mice. Three-month-old TS mice (*n* = 6/group) were orally administered with GFB (8 mg/kg/d) solubilized in 0.5% methylcellulose. Therefore, control TS mice received 0.5% methylcellulose as a vehicle. After 2 months of treatment, cortical sections were stained for TUNEL (**A**). TUNEL (+ve) cells were expressed as a % of total cells per square mm (**B**). Paraffin-embedded cerebral cortex sections were subjected to periodic acid-Schiff (PAS) stain for analyzing glycolipids (**C**). The glycoconjugate material-stained magenta is shown by the arrow and quantified for PAS +ve cells per mm square (**D**). Paraffin-embedded sections were also subjected to H & E staining as characterized by large vacuoles (thick arrows), pyknotic nuclei and swollen neuron in TS mice (**E**). The number of vacuoles was quantified per mm square (**F**). Results represent counting three different sections from 3 different mice (*n* = 3) per group. All data represent mean ± SEM. All statistical analysis was performed by one-way ANOVA, followed by Dunnett’s multiple comparison test. *** *p* < 0.001.

**Figure 5 cells-12-02791-f005:**
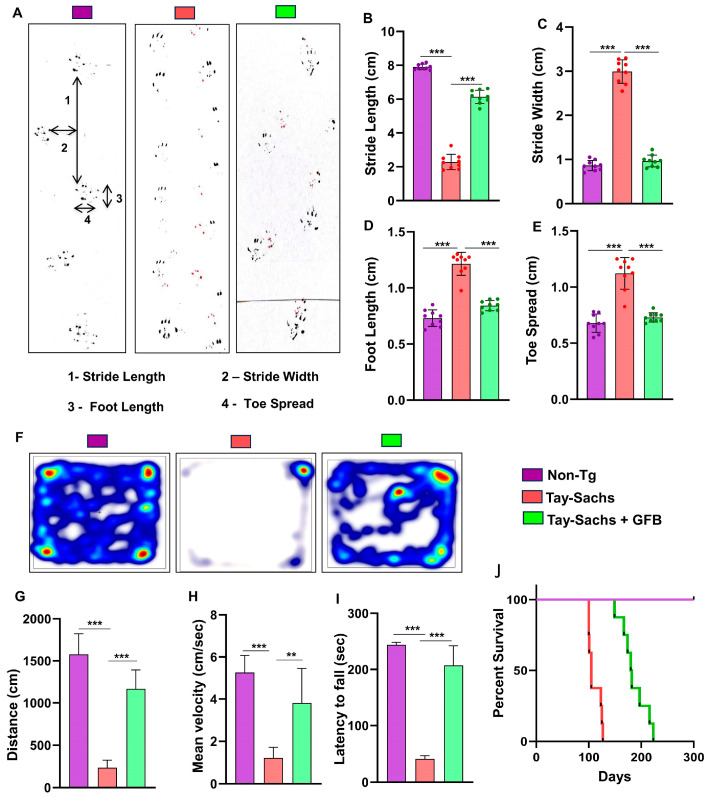
Oral administration of GFB improves gait, alleviates motor deficits, and increases survival in Tay-Sachs mice. Gait analysis, open field locomotor activities, and rotarod performance were performed on three-month-old TS mice (*n* = 6) treated with GFB (8 mg/Kg/day) via oral gavage for two months. Mice painted with different colors on front (red) and hind paws (black) were allowed to walk through a clear tunnel. After pre-training sessions, the footprints were measured on a white paper on the runway floor. Footprints were measured for stride length (**A**,**B**), stride width (**A**,**C**), foot length (**A**,**D**), and toe spread (**A**,**E**). Heatmaps demonstrate the horizontal locomotor activities of experimental animals in the open field arena as captured by the Noldus software (**F**). Parameters related to the movement of animals were obtained from the software and presented as the total distance moved for all the groups of mice (**G**), mean velocity (**H**). Rotarod test exhibiting the feet movement of animals on the rotating rod and the latency time taken by each mouse to fall on the base were monitored (**I**). Kaplan–Meier survival analysis of WT (n = 8), Hex^−/−^ (*n* = 8), and Hex^−/−/^GFB (*n* = 8) (**J**). All data represent mean ± SEM. All statistical analyses were performed by one-way ANOVA, followed by Dunnett’s multiple comparison test; ** *p* < 0.01; *** *p* < 0.001. ns—non-significant. Data are represented as mean ± SEM.

**Figure 6 cells-12-02791-f006:**
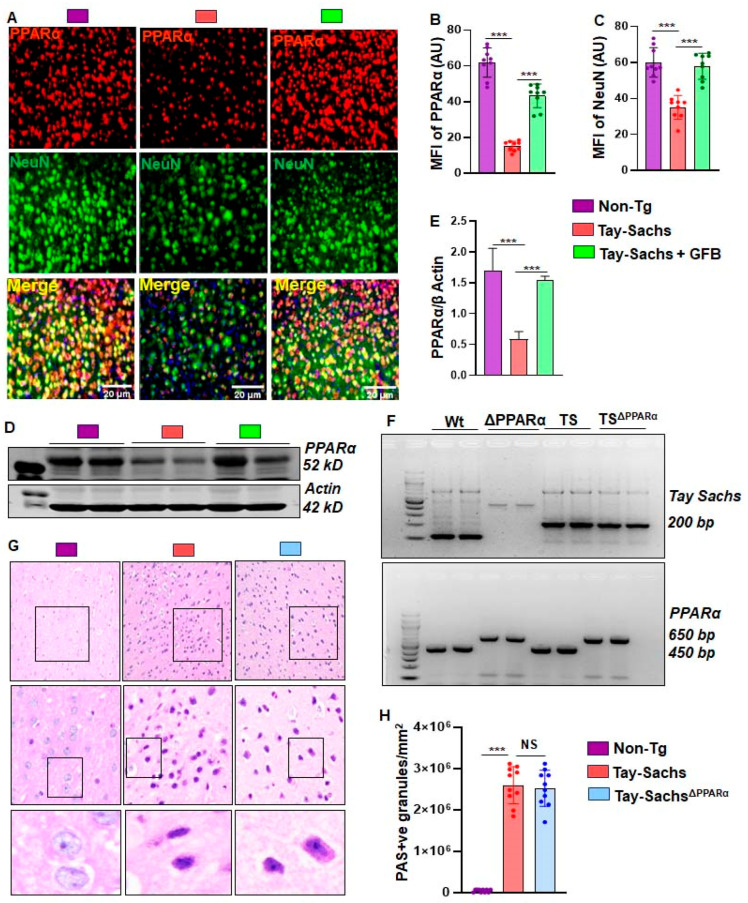
Role of PPARα in the accumulation of glycoconjugates in the brain of Tay-Sachs mice. Representative immunohistochemical images of the motor cortex region of Tay-Sachs, Tay-Sachs + GFB, and Non-Tg mice compared and co-stained for overall PPARα (red stain) and NeuN (green stain) expression. PPARα and NeuN were quantified from a total of nine images (20× magnification) from 3 different mice per group (**A**–**C**). PPARα protein expression levels were compared between the same groups using immunoblot analysis (**D**), followed by densitometric quantification of the same (**E**). TS mice bred with PPARα knockout mice to generate Tay-Sachs^ΔPPARα^ lines. Representative PCR of Tay-Sachs and PPARα in 6-month-old Non-Tg, Tay-Sachs, and Tay-Sachs^ΔPPARα^ mice (**F**). Representative PAS (Periodic acid stain) to study the expression of glycoconjugate levels in Non-Tg, Tay-Sachs, and Tay-Sachs^ΔPPARα^ (**G**). The glycoconjugate material stained magenta is quantified for PAS +ve cells per mm square (**H**). Results represent counting three different sections from 3 different mice (*n* = 3) per group. All data represent mean ± SEM. All statistical analyses were performed by one-way ANOVA, followed by Dunnett’s multiple comparison test. *** *p* < 0.001.

**Figure 7 cells-12-02791-f007:**
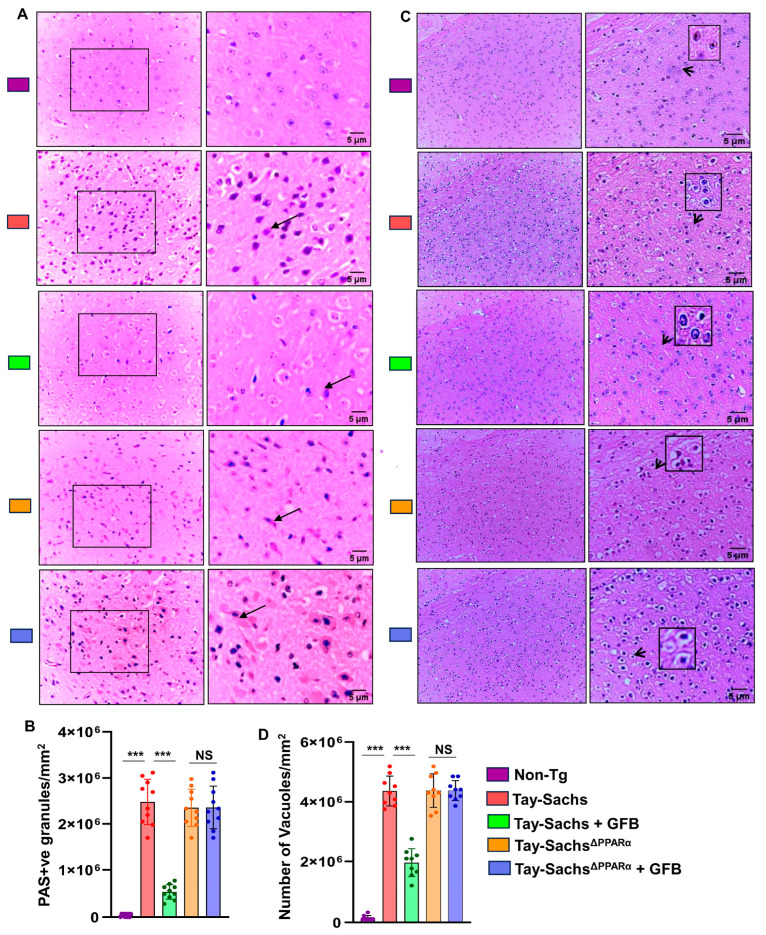
Oral administration of GFB reduces glycoconjugate storage in Tay-Sachs mice via PPARα. Tay-Sachs mice (*n* = 6/group) of three months age were treated with GFB (8 mg/kg/d) via oral gavage. After 2 months of daily treatment, paraffin-embedded cerebral cortex sections were subjected to periodic acid-Schiff (PAS) stain for analyzing glycolipids. The glycoconjugate material stain magenta is shown by the thin arrows (**A**). The glycoconjugate material-stained magenta is shown by the arrow and quantified for PAS +ve cells per mm square (**B**). Paraffin-embedded sections were also subjected to H & E staining, as characterized by large vacuoles (thick arrows), pyknotic nuclei, and swollen neuron in Tay-Sachs mice (**C**). The number of vacuoles was quantified per mm square (**D**). Results represent counting three different sections from 3 different mice (*n* = 3) per group. All data represent mean ± SEM. All statistical analyses were performed by one-way ANOVA, followed by Dunnett’s multiple comparison test. *** *p* < 0.001.

**Figure 8 cells-12-02791-f008:**
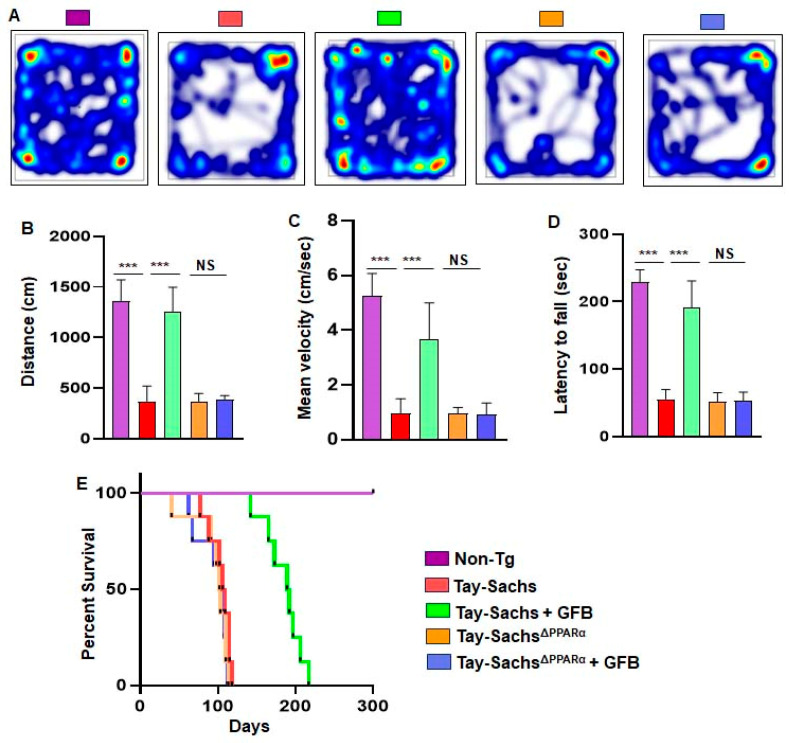
Oral administration of gemfibrozil improves gait, alleviates motor deficits, and increases survival in Tay-Sachs mice via PPARα. Open field locomotor activities and the rotarod test were performed on three-month-old Tay-Sachs mice (*n* = 6) and Tay-Sachs^ΔPPARα^ treated with gemfibrozil (8 mg/Kg/day) via oral gavage for two months. Heatmaps demonstrate the horizontal locomotor activities of experimental animals in the open field arena as captured by the Noldus software (**A**). Parameters related to movement of animals were obtained from the software and presented as the total distance moved for all the groups of mice (**B**), mean velocity (**C**). Rotarod test exhibiting the feet movement of animals on the rotating rod and the latency time taken by each mouse to fall on the base were monitored (**D**). Kaplan–Meier survival analysis of Non-Tg (*n* = 8), Tay-Sachs (*n* = 8), Tay-Sachs+GFB (*n* = 8), Tay-Sachs^ΔPPARα^ (*n* = 8), and Tay-Sachs^ΔPPARα^ + GFB (*n* = 8) (**E**). All data represent mean ± SEM. All statistical analyses were performed by one-way ANOVA, followed by Dunnett’s multiple comparison test; *** *p* < 0.001. ns—nonsignificant. Data are represented as mean ± SEM.

## Data Availability

The data presented in this study is included in the main text.

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
