# Peer review of "Lipid-Lowering Drug Gemfibrozil Protects Mice from Tay-Sachs Disease via Peroxisome Proliferator-Activated Receptor α"

_cells, 2023, doi:10.3390/cells12242791_

Round 1

Reviewer 1 Report

Comments and Suggestions for Authors

The authors have conducted a double transgenic mouse studies to implicate PPARalpha as a critical molecular pathway by which GFB (Lopid) exerts its therapeutic effects in a Tay-Sachs disease mouse model.

While they have provided biochemical and functional evidence to show that PPARalpha activation will be an important target for therapeutic intervention for Tay Sachs by GFP, there is some key background information that needs to be included.

Major

1.  Authors should discuss literature that describes the therapeutic effect of PPAR-alpha activation in other mouse models and human neuronal models of neurodegenerative diseases.  

2.  Please also discuss key literature cited in this review that was recently published "Neuroprotective effects of gemfibrozil in neurological disorders: Focus on inflammation and molecular mechanisms." by Ivraghi et al., 2023.  In particular Gottschalk et al., have already shown that GFB can protect dopaminergic neurons in a mouse model of PD through a PPARa-dependent astrocytic GDNF signal. These findings are extremely relevant to the authors' interest in application of GFB in Tay-Sachs.  The molecular mechanism and applicability of GFB treatment has been shown before in other neurodegenerative models reducing the novelty of the current study.  Although, the current study is novel in the sense of application to Tay-Sachs in particular so authors should highlight that while describing other's findings in the background in more detail.

3.  In the introduction, authors should mention what the incidence of Tay- Sachs disease is.

4.  In the introduction, authors should elaborate more clearly on what the cellular endogenous function of HEXA is and what are the types of mutations that cause Tay-Sachs; what do the mutations do to the HEXA protein (ex. null mutations? loss of expression or loss of function? or both types of mutations?).

Minor

Line 422: Should say "...treatment of hyperlipidemia..."

Line 472: What is the full form of LINCL?

Author Response

Major comments

1.  Authors should discuss literature that describes the therapeutic effect of PPAR-alpha activation in other mouse models and human neuronal models of neurodegenerative diseases.

Response: We have done it. Please see lines 490 to 503. Thanks.  

2.  Please also discuss key literature cited in this review that was recently published "Neuroprotective effects of gemfibrozil in neurological disorders: Focus on inflammation and molecular mechanisms." by Ivraghi et al., 2023.  In particular Gottschalk et al., have already shown that GFB can protect dopaminergic neurons in a mouse model of PD through a PPARa-dependent astrocytic GDNF signal. These findings are extremely relevant to the authors' interest in application of GFB in Tay-Sachs.  The molecular mechanism and applicability of GFB treatment has been shown before in other neurodegenerative models reducing the novelty of the current study.  Although, the current study is novel in the sense of application to Tay-Sachs in particular so authors should highlight that while describing other's findings in the background in more detail.

Response: We have done it. Please see lines 490 to 503. Thanks.  

3.  In the introduction, authors should mention what the incidence of Tay- Sachs disease is.

Response: We have done it. Please see lines 33 to 36. Thanks.   

4.  In the introduction, authors should elaborate more clearly on what the cellular endogenous function of HEXA is and what are the types of mutations that cause Tay-Sachs; what do the mutations do to the HEXA protein (ex. null mutations? loss of expression or loss of function? or both types of mutations?).

Response: We have done this. Please see lines 38 to 45.  

Minor comments

Line 422: Should say "...treatment of hyperlipidemia..."

Response: We have fixed it. Thanks.  

Line 472: What is the full form of LINCL?

Response: We have done this. Please see line 490. Thanks.  

Reviewer 2 Report

Comments and Suggestions for Authors

1.      Please show the chemical structure of gemfibrozil (GFB).

2.      Please fix some typos.

3.      What are NTg and TS samples?

4.      Based on quick search, one of the side effects of GFB is increase cancer risk. Can it affect the apoptosis of cancer cells as well? Or, during the experiments, did the authors observe the occurrence of cancer from the mice?

5.      The authors should explain why they use GFB in this study among various drugs in detail.

6.      Please reorganize the location of figures. If possible, please put the figures in the same page where they are cited.

Author Response

  1. Please show the chemical structure of gemfibrozil (GFB).

Response: We have done it. Please see Figure 1. Thanks.    

  1. Please fix some typos.

Response: We have done it. Thanks.  

  1. What are NTg and TS samples?

Response: We have mentioned these clearly in Figures and text. Thanks.  

  1. Based on quick search, one of the side effects of GFB is increase cancer risk. Can it affect the apoptosis of cancer cells as well? Or, during the experiments, did the authors observe the occurrence of cancer from the mice?

Response: We have mentioned this. Please see lines 437-440. Thanks.  

  1. The authors should explain why they use GFB in this study among various drugs in detail.

Response: As compared to other fibrate drugs, GFB has a better safety profile. Since FDA approval in 1981, this drug is being continuously used without any major side effect. Therefore, we selected GFB to other fibrate drugs.  

  1. Please reorganize the location of figures. If possible, please put the figures in the same page where they are cited.

Response: We have done that. Thanks.